# Influences of Breathing Exercises and Breathing Exercise Combined with Aerobic Exercise on Changes in Basic Spirometry Parameters in Patients with Bronchial Asthma

**Ľuboš Grznár** [1,*]**, Dávid Sucháň** [1]**, Jana Labudová** [1]**, Lukáš Odráška** [2] **and Ivan Matúš** [3]

[1] Department of Outdoor Sports and Swimming, Faculty of Physical Education and Sport, Comenius University, 814 69 Bratislava, Slovakia; suchan9@uniba.sk (D.S.); jana.labudova@uniba.sk (J.L.)

[2] Department of School Education, Faculty of Education, Trnava University, 918 43 Trnava, Slovakia; lukas.odraska@gmail.com

[3] Department of Educology of Sports, Faculty of Sports, University of Presov, 080 01 Prešov, Slovakia; ivan.matus@unipo.sk

[*] Correspondence: lubos.grznar@uniba.sk; Tel.: +421-2-90179936

**Abstract:** Scientific evidence shows that breathing or aerobic programs can improve the quality of life of asthma patients. The aim of this work was to find out the influences of breathing exercises and breathing exercises combined with aerobic exercise on changes in spirometry parameters in patients with bronchial asthma. Participants: The group consisted of 33 women with bronchial asthma—mild to moderate persistent levels of FEV1 reduction (80–50%)—with a mean age of 34.73 ± 1.53 years. They were randomly assigned to experimental group 1 (EX1), experimental group 2 (EX2) or the control group (CG). Materials and methods: Changes in spirometry parameters were evaluated over a 16-week period in the three groups: CG (placebo), EX1 (breathing exercises) and EX 2 (combination of breathing exercises with an aerobic program). To evaluate the pre-training and post-training diagnostics, we used MIR Spirobank II. The influences of the experimental and control factors were assessed using the following dependent variables: forced vital capacity (FVC), forced expiratory volume in one second (FEV1), Tiffeneau–Pinelli index (FEV1/FVC ratio), peak expiratory flow (PEF) and forced mid-expiratory flow (FEF25–75%). We used the Wilcoxon t-test and the Kruskal–Wallis test to evaluate the differences in the measured parameters. To examine the effect of our protocols, we used effect size (ES). Results: In CG we observed improvements in: FVC—(5%; $p < 0.05$; ES = 0.437). FEV1—(7.33%; $p < 0.01$; ES = 0.585). FEV1/FVC ratio (5.27%; $p < 0.01$; ES = 0.570). PEF (11.22%; $p < 0.01$; ES = 0.448). FEF25–75% (7.02%; $p < 0.01$; ES = 0.628). In EX1 we observed improvements in: FVC (5.23%; $p < 0.01$; ES = 0.631), FEV1 (20.67%; $p < 0.01$; ES = 0.627), FEV1/FVC ratio (16.06%; $p < 0.01$; ES = 0.628), PEF (13.35%; $p < 0.01$; ES = 0.627) and FEF25–75% (13.75%; $p < 0.01$; ES = 0.607). In EX2 we observed improvements in: FVC (9.12%; $p < 0.01$; ES = 0.627), FEV1 (27.37%; $p < 0.01$; ES = 0.626), FEV1/FVC ratio (15.32%; $p < 0.01$; ES = 0.610), PEF (30.66%; $p < 0.01$; ES = 0.626) and FEF25–75% (58.99%; $p < 0.01$; ES = 0.626). Significant differences compared to the control group were observed in EX1 for FEV1 ($p < 0.05$) and FEV1/FVC ratio ($p < 0.01$); and in EX2 for FEV1 ($p < 0.05$), FEV1/FVC ratio ($p < 0.01$), PEF ($p < 0.05$) and FEF ($p < 0.05$). A significant difference between EX1 and EX2 was observed in PEF ($p < 0.05$). Conclusions: It appears to be that combination of breathing exercises with aerobic activities is a more beneficial option for patients with bronchial asthma.

**Keywords:** bronchial asthma; breathing exercises; aerobic activity; spirometry

## 1. Introduction

Asthma affected approximately 262,000,000 people in 2019 and caused 461,000 deaths [1]. In adults, the overall prevalence of diagnosis is estimated to be 4.3% [2].

Pharmacological treatment has been shown to significantly improve symptoms of asthma [3–5]. The use of inhaled corticosteroids, long acting β2-agonists and combinations

of these medications have been shown to improve asthma control [3,4], but compliance rates of >80% are required to maintain this level of control [6]. Compliance with asthma treatment in countries where treatment is readily accessible remains poor [7,8].

Asthma control is determined by the frequency of daytime symptoms, limitation of activities, nocturnal symptoms, need for redelivering medication, lung function and exacerbations [1]. Patients are classified as having controlled, partially controlled or uncontrolled asthma. Data show that only 23% of asthmatics are controlled [9], and despite receiving specialist care, 50% are not well controlled [8]. Poor asthma control has been associated with more emergency room visits, physician visits and days spent in hospital [10].

A program of breathing exercises was found to be effective and could be completed in less than 10 min per day. Furthermore, there was a significant improvement in Asthma Control Test scores post-training [11]. The most common breathing exercise are diaphragm breathing, yoga Breathing and the Buteyko method. Breathing methods used to increase diaphragm breathing involve improving the strength and endurance of the respiratory muscles [12]. Training of diaphragm breathing is focused on increase the inspiratory capacity, which is based on the principle of overloading and applies a load to the supporting muscles of the diaphragm and inspiratory, respiratory muscles [13]. A training program using diaphragm resistive breathing is an intervention for endurance and strength because a weakening of the inspiratory muscle interferes with the motor performance and breathing skills [14]. The method using yoga breathing may include a period of breath holding following either inhaling or exhaling. Hence, most simple techniques are a way to voluntarily slow down and prolong breathing. The correct or most used technique for breathing according to yoga is recognized to be slow and deep with inhalation and exhalation in a ratio of 1:2 [15]. The major component of the Buteyko method is to reduce hyperventilation through periods of controlled reduction in breathing, known as "slow breathing" and "reduced breathing", combined with periods of breath holding, known as "control pauses" and "extended pauses". These techniques are very similar to those routinely used by respiratory physiotherapists for patients with hyperventilation symptoms [16,17]. Patients with moderate-to-severe asthma who participated in aerobic programs or breathing programs presented similar results in asthma control, quality of life, asthma symptoms, psychological distress, physical activity and airway inflammation. However, a greater proportion of participants in the aerobic training group achieved improvements in asthma control and reduced use of rescue medication [18]. Recent research demonstrates that healthcare use is higher in physically inactive asthmatics compared with physically active asthmatics [19]. This finding suggests that active asthmatics have better asthma control if healthcare use is a proxy for asthma control. Exercise interventions involving adults with asthma have shown improvements in measures such as lung function [20], quality of life [21], breathlessness [21–23] and controller therapy [24]; and animal models have shown improvements in airway inflammation [25,26]. The effect of breathing exercise combined with aerobic exercise was found to be effective, and in patients with asthma, improvements in FVC, $FEV_1$ and $VO_2$max were observed [27,28]. However, a direct association between asthma control and exercise program has not yet been obtained. The intensity used for the various aerobic load protocols has a large variance: from 50% to 80% of HR max. An intensity of 80% was used in the studies [29–32]. The frequency of stimuli per week, in supportive treatment of asthma, is usually two [33–35] or three [36,37] times. In a meta-analysis [38] in 2020, the authors concluded that continuous aerobic activity at moderate intensity for at least 20 min two or three times per week for at least four weeks is suitable for non-pharmacological supportive treatment of asthma.

The aim of our work was to determine the effects of breathing exercise and breathing exercise combined with aerobic exercise on changes in individual parameters of spirometry examination in patients with bronchial asthma.

## 2. Materials and Methods

This study was conducted in the Department of Allergy and Clinical Immunology at St. Svorad Hospital, Nitra, Slovakia, in collaboration with The Department of Outdoor Sports and Swimming, Comenius University, Bratislava, Slovakia.

### 2.1. Participants

Each selected subject was acquainted in advance with the content, goal and conditions of the research work by email, and after, subsequent consent was obtained in the research. The initial briefing for the research took place through online meetings, where we presented the factors we used to improve their condition. During the experimental period, patients were asked to follow the treatment and the doctor's instructions, and to maintain their usual diet and exercise regimes. Selection criteria: 1. Adult women 40 > 30 years of age. 2. A mildly decreased FEV1 at the level of 80 to 50% compared to the reference values. 3. Individuals who did not engage in sport at national or top level in the past or at the time of our work. Exclusion criteria: 1. Acute illness (confirmed by a doctor) that would limit the ability of the patient to participate in the study. 2. Participation less than 90%. 3. Refusal to give informed consent. Participants were women ($n = 33$), $34.73 \pm 1.53$ years, who participated voluntarily at this study. Participants were randomly assigned into three groups. At the end of the experiment, due to exclusion criteria, numbers of participants in groups: experimental group 1 (EX1), $n = 10$—breathing exercise group; experimental group 2 (EX2), $n = 11$—breathing exercise + aerobic program; control group (CG), $n = 12$, placebo. Initial parameters of participants in groups are given in Table 1. There were not significant ($p > 0.05$) differences between groups at baseline.

**Table 1.** Initial parameters.

| Parameters | CG | EX1 | EX2 |
|---|---|---|---|
| Participants [$n$] | 12 | 10 | 11 |
| Age | $34.80 \pm 1.93$ | $34.82 \pm 1.33$ | $34.58 \pm 1.44$ |
| Height (cm) | $166.08 \pm 4.25$ | $166.10 \pm 4.89$ | $167.27 \pm 4.54$ |
| Weight (kg) | $66.33 \pm 4.33$ | $69.50 \pm 6.29$ | $68.91 \pm 4.97$ |
| FVC (L) | $2.95 \pm 0.21$ | $2.87 \pm 0.23$ | $2.96 \pm 0.25$ |
| FEV1 (L) | $1.91 \pm 0.23$ | $1.79 \pm 0.28$ | $1.79 \pm 0.20$ |
| FEV1/FVC (%) | $74.33 \pm 6.51$ | $74.1 \pm 7.56$ | $73.55 \pm 5.84$ |
| PEF (L/s) | $4.1 \pm 1.03$ | $4.27 \pm 0.48$ | $4.37 \pm 0.9$ |
| FEF25–75% (L/s) | $1.71 \pm 0.59$ | $1.6 \pm 0.45$ | $1.39 \pm 0.35$ |

### 2.2. Intervention and Instruments

2.2.1. Experimental Protocol 1

Probands performed breathing exercises 4 times a week (Monday, Wednesday, Friday, Sunday) on an empty stomach in the morning for 16 weeks. The breathing exercise program lasted a total of 20 min. The total number of breathing exercises was 64. Patients performed four breathing methods—diaphragm breathing, the Buteyko method, resistance breathing and yoga breathing. Due to COVID-19 restrictions, probands performed exercises home under the supervision of coach via online video platform. The descriptions of breathing exercises are stated in the Table 2.

**Table 2.** Descriptions of breathing exercises.

| Breathing Exercises | Breathing Cycles [*n*] | Series [*n*] | Method |
|---|---|---|---|
| Balloon inflation | 10 | 2 | Resistance |
| Diaphragmatic breathing while standing | 6 | 2 | Diaphragm |
| Diaphragmatic breathing while sitting | 5 | 2 | Diaphragm |
| Diaphragmatic breathing in bed with legs bent | 4 | 3 | Diaphragm |
| Torso rotation with active breathing in the sideways | 10 | 2 | Yoga |
| Squat with spine rotation and active breathing | 10 | 2 | Yoga |
| Walking with alternating rhythm of breathing | 12 | 2 | Buteyko |
| Running with alternating rhythm of breathing | 12 | 4 | Buteyko |
| Breath holding | 6 | 4 | Buteyko |

2.2.2. Experimental Protocol 2

This program consisted of breathing exercises and an aerobic program. Probands performed breathing exercises with the same conditions, times and amounts as patients in the breathing exercises only program. The aerobic program depended on the current physical condition of the probands and consisted of run or walk. Probands were instructed to perform physical activity at vigorous intensity. Intensity was set for every proband individually at 80% of her maximal hart rate. Maximal heart rate was calculated as result of the formula $220 - \text{age}$. Probands were instructed to maintain 80% of HR max ($\pm 2$ BPM) and not to exceed this threshold. Intensity parameters were checked with a heart rate monitor. After each training session, the probands sent the course of the training load to the supervisor to control. Probands, after 10 min of warming up (110–120 BPM), started to exercise at the set intensity. For each proband, the exercise was chosen individually according to his current physical condition. Probands could choose running or Nordic walking. The exercise was completed after 30 min, which they spent at 80% HR max. After the main part, 5 min of easy walking (100–110 BPM) followed for body recovery. Probands performed aerobic activity 3 times per week (Tuesday, Thursday, Saturday) for 30 min for 16 weeks.

2.2.3. Control Protocol

Probands in this protocol were instructed to perform their normal activities during the day as usual. This program lasted for 16 weeks, like the experimental ones. As a part of the follow-up, we gave to patients a placebo program in the form of an inhaler, which had no significant effects on improving respiratory problems. Patients were instructed to take the three recommended doses daily (morning, lunch, evening) and to maintain their physical activity at approximately the same level during the control period and not to change their diet significantly.

*2.3. Selection Criteria*

We monitored strenuous expiratory parameters (FVC—forced vital capacity; FEV1—volume expired in the 1st second of the test; FEV1/FVC − FEV1/FVC × 100 (Tiffeneau–Pinelli index); PEF—peak expiratory flow; FEF25–75%—average flow between 25% and 75% of the FVC) in the standard spirometry examination procedure.

*2.4. Data Collections*

The pre-training diagnosis was performed in the week 28 September 2020–2 October 2020, and post-training diagnostics were performed in the week 25 January 2021–29 January 2021. To obtain empirical data, we used diagnostics using spirometry examination, using the device SPIROBANK II (MIR SRL, Rome, Italy). Spirometry specifications of this device were

volume accuracy $\pm$ 2.5% or 50 mL; flow accuracy $\pm$ 5% or 200 mL/s; flow range $\pm$ 16 L/s; dynamic resistance < 0.5 cm $H_2O$/L/s. Before each examination, we performed a calibration check following the recommendations of the experts and the instructions of the spirometer manufacturer. This control consisted of performing three cycles with a calibration pump with a volume of 3 L and a flow rate of 0.5–12 L/s, and with minimum range of $\pm$15 mL. Reliability was validated by a commonly performed biological test-repeated examinations by workplace experts and comparison of compliance with previous examinations.

### 2.5. Data Analysis

We used IBM SPSS STATISTICS 25 (IBM, Armonk, NY, USA) to process and evaluate the data we obtained. We used a nonparametric Wilcoxon *t*-test for paired sets to determine statistical significance in individual groups separately between pre-training and post-training diagnostics. We used a nonparametric Kruskal–Wallis test to determine the statistical significance between the post-training values of the individual groups. To measure the practical significance and the effect of the methods used, according to [39,40], the effect coefficient "r"—effect size (ES)—was calculated. Differences were considered statistically significant if $p < 0.05$.

## 3. Results

### 3.1. Forced Vital Capacity

The following changes were observed (Figure 1): The CG group improved from 2.95 $\pm$ 0.21 [L] to 3.05 $\pm$ 0.21 [L]; 3.39%; $p < 0.05$; ES = 0.437. The EX1 group improved from 2.87 $\pm$ 0.23 [L] to 3.02 $\pm$ 0.21 [L]; 5.23%; $p < 0.01$; ES = 0.631. The improvement in the EX2 group was from 2.96 $\pm$ 0.25 [L] to 3.23 $\pm$ 0.16 [L]; 9.12%; $p < 0.01$; ES = 0.627. Post-training values between groups were not significant ($p > 0.05$).

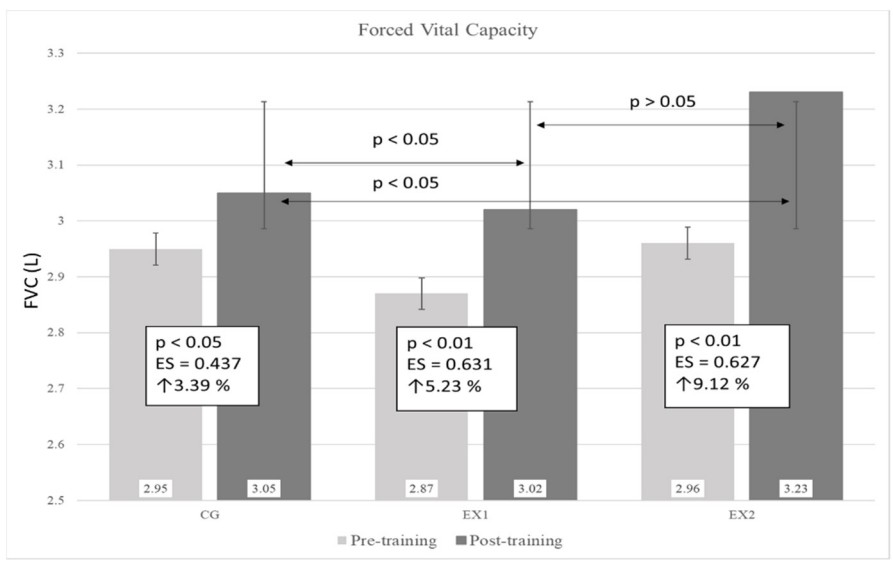

**Figure 1.** Force vital capacity.

### 3.2. Volume Expired in the 1st Second of the Test

The following changes was observed (Figure 2): The CG group improved from 1.91 $\pm$ 0.23 [L] to 2.05 $\pm$ 0.19 [L]; 7.33%; $p < 0.01$; ES = 0.585. The EX1 group improved from 1.79 $\pm$ 0.28 [L] to 2.16 $\pm$ 0.22 [L]; 20.67%; $p < 0.01$; ES = 0.627. The improvement in the EX2 group was from 1.79 $\pm$ 0.20 [L] to 2.28 $\pm$ 0.21 [L]; 27.37%; $p < 0.01$; ES = 0.626. Post-training values between EX1 and CG were significant ($p < 0.05$). Post-training values between EX2 and CG were significant ($p < 0.05$). Post-training values between EX1 and EX2 were not significant ($p > 0.05$).

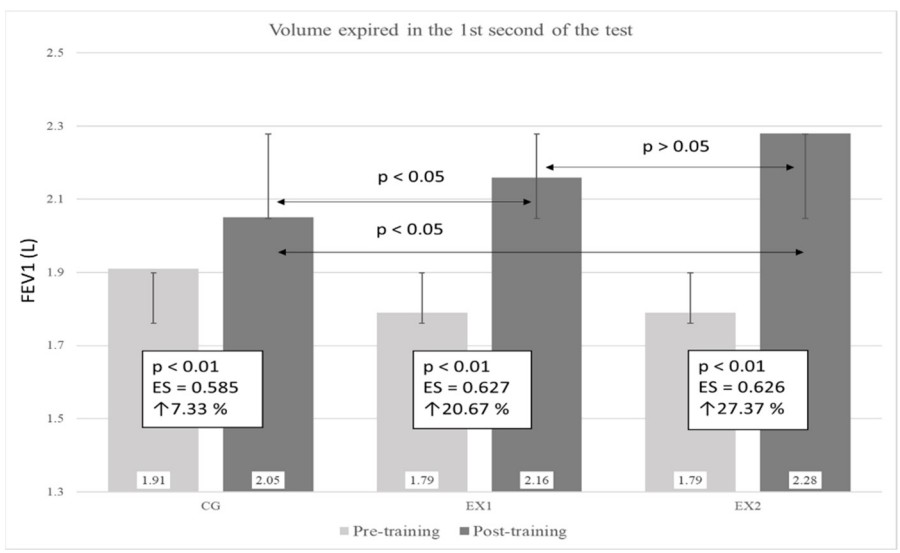

**Figure 2.** Volume expired in the first second of the test.

### 3.3. Tiffeneau–Pinelli Index

The following changes was observed (Figure 3): The CG group improved from 74.33 ± 6.51% to 78.25 ± 5.34%; 5.27%; $p < 0.01$; ES = 0.570. The EX1 group improved from 74.1 ± 7.56% to 86 ± 6.15%; 16.06%; $p < 0.01$; ES = 0.628. The improvement in the EX2 group was from 73.55 ± 5.84% to 84.82 ± 3.95%; 15.32%; $p < 0.01$; ES = 0.610. Post-training values between EX1 and CG were significant ($p < 0.05$). Post-training values between EX2 and CG were significant ($p < 0.05$). Post-training values between EX1 and EX2 were not significant ($p > 0.05$).

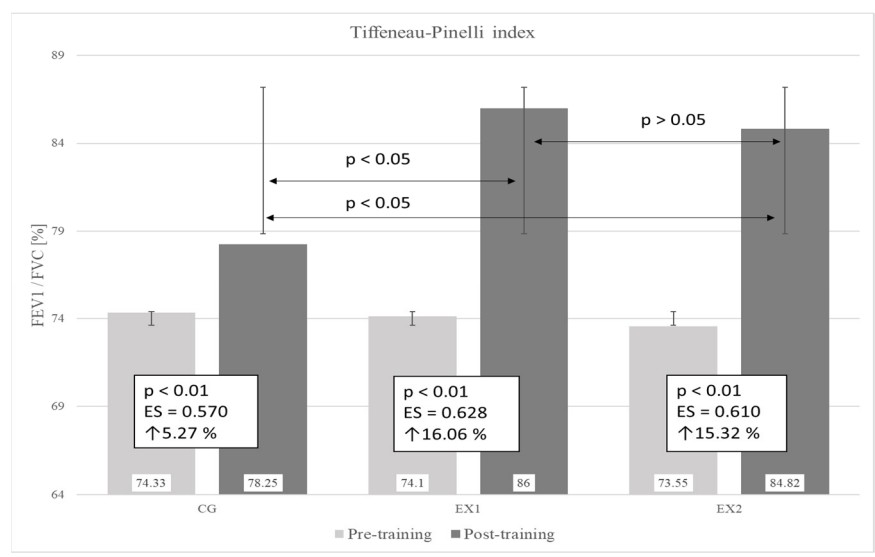

**Figure 3.** Tiffeneau–Pinelli index.

### 3.4. Peak Expiratory Flow

The following changes were observed (Figure 4): The CG group improved from 4.1 ± 1.03 [L/s] to 4.56 ± 0.81 [L/s]; 11.22%; $p < 0.05$; ES = 0.448. The EX1 group improved from 4.27 ± 0.48 [L/s] to 4.84 ± 0.55 [L/s]; 13.35%; $p < 0.01$; ES = 0.627. The improvement in the EX2 group was from 4.37 ± 0.9 [L/s] to 5.71 ± 0.31 [L/s]; 30.66%; $p < 0.01$; ES = 0.610. Post-training values between groups were significant ($p < 0.05$) between EX2 and CG. Post-training values between EX1 and CG were not significant ($p > 0.05$). Post-training values between EX1 and EX2 were not significant ($p > 0.05$).

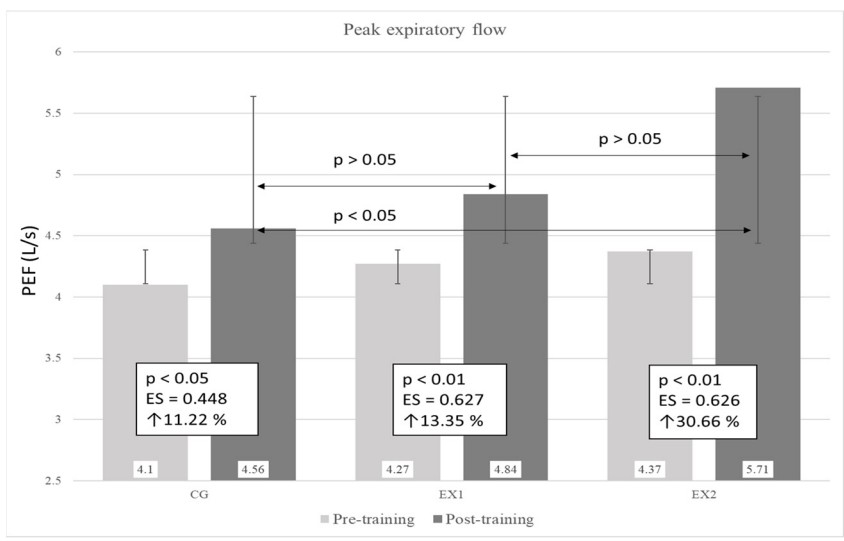

**Figure 4.** Peak expiratory flow.

*3.5. Average Flow between 25% and 75% of the FVC*

The following changes was observed (Figure 5): The CG group improved from $1.71 \pm 0.59$ [L/s] to $1.83 \pm 0.49$ [L/s]; 7.02%; $p < 0.01$; ES = 0.628. The EX1 group improved from $1.6 \pm 0.45$ [L/s] to $1.82 \pm 0.38$ [L/s]; 13.75%; $p < 0.01$; ES = 0.607. The improvement in the EX2 group was from $1.39 \pm 0.35$ [L/s] to $2.21 \pm 0.39$ [L/s]; 58.99%; $p < 0.01$; ES = 0.626. Post-training values between groups were significant ($p < 0.05$) between EX2 and CG. Post-training values between EX1 and CG were not significant ($p > 0.05$). Post-training values between EX1 and EX2 were not significant ($p > 0.05$).

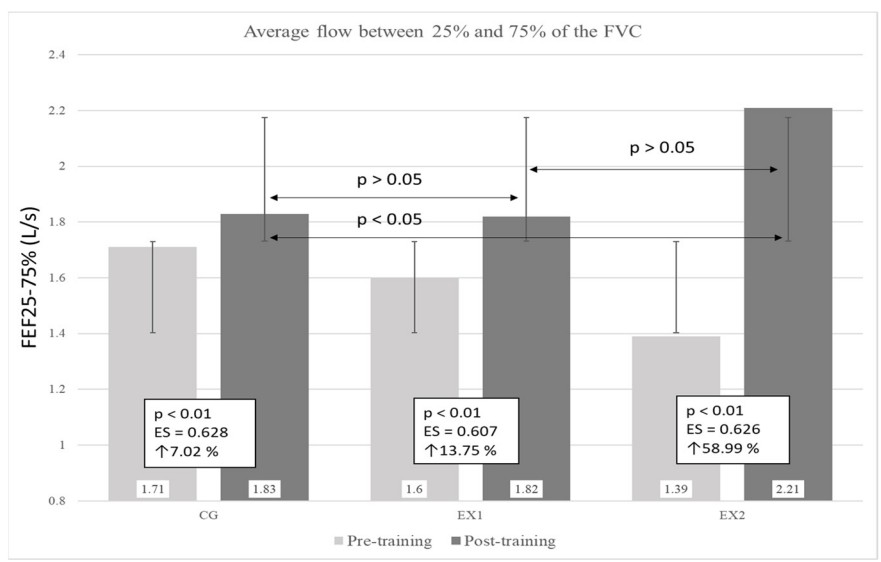

**Figure 5.** Average flow between 25% and 75% of the FVC.

## 4. Discussion

Meta-analysis from 2020 concluded that breathing exercises may have positive effects on quality of life, hyperventilation symptoms and lung function in adults with mild to moderate asthma [41]. The aim of the study was to investigate the effects of breathing exercises and a combination of breathing exercises and an aerobic program on changes in basic spirometry parameters in standardized diagnostics in patients with bronchial asthma in the presence of a clinical immunologist.

The British Thoracic Society guideline on rehabilitation of adult pulmonary functions suggests that people with chronic respiratory disorders should undergo respiratory rehabilitation and points out that routine exercise in asthma patients is not recommended based on inadequate clinical evidence [42]. Asthmatics are often careful when they hear about exercises that cause bronchoconstriction [43–46], which is why some of them prefer a sedentary lifestyle and try to avoid physical activity and prevent exacerbations [29–47].

Our results showed in experimental group 1 (breathing exercises) significant improvements in two parameters (FEV1 (20.67%) and FEV1/FVC (16.06%)); and in experimental group 2 (aerobic program and breathing exercises), we found significant improvements in up to four parameters of spirometry examination (FEV1 (27.37%); FEV1/FVC (15.32%); PEF (30.66%); FEF25–75 (58.99%)) compared to the control group. Both experimental programs showed more effective improvements than the control group. A closer comparison of the experimental groups showed higher percentage increments for EX2 compared to EX1. The combination of breathing exercises with an aerobic program appeared to be a more effective means of improvement at the level of spirometry examination than others. However, among CG, EX1 and EX2, there was no significant difference. The effectiveness of aerobic exercise as a supportive treatment for asthma has also been confirmed by [37], which concluded that aerobic training reduced bronchial hyperresponsiveness and serumproinflammatory cytokines, and improved quality of life and asthma exacerbation in patients with moderate or severe asthma. These results suggest that adding exercise as an adjunct therapy to pharmacological treatment could improve the main features of asthma. Meta-analyses [48] which summarized effect of exercise-based pulmonary rehabilitation on adults with asthma found no significant improvement (MD = 0.10, 95% CI: −0.08 to 0.29) in forced expiratory volume in 1 s. Nonetheless, improvements in forced vital capacity (MD = 0.23, 95% CI: 0.08 to 0.38) and peak expiratory flow (MD = 0.39, 95% CI: 0.21 to 0.57) were significant. Effectiveness of breathing exercise was the main objective of this study [49]. The outcome measurement tools that we used were the Asthma Control Test (ACT), Asthma Quality of Life Questionnaire (AQLQ), pulmonary function test (PFT), and the patient observation chart. Results show that the breathing exercise group had a significantly higher average ACT score, overall AQLQ score, and subscale scores than the relaxation group ($p < 0.05$). However, there was no significant difference between the groups in terms of PFT parameters and peak expiratory flow values ($p > 0.05$). We concluded that breathing exercise improved asthma control and asthma-related quality of life in people with asthma, but it did not show a significant difference in PFT values. Meta-analysis [41] which summarized effect of breathing exercises for adults with asthma found in lung functions low evidence of effectiveness breathing programs. Forced expiratory volume in 1 s measured at up to three months was inconclusive, MD −0.10 L, (95% CI: −0.32 to 0.12; 4 studies, 252 participants, very low-certainty evidence). However, for FEV1% predicted, an improvement was observed in favor of the breathing exercise group (MD 6.88%, 95% CI: 5.03 to 8.73; five studies, 618 participants). Comparation of breathing and aerobic exercises was realized by [50], which concluded that the breathing exercise intervention was effective at improving the lung volumes in asthmatic children. The aerobic exercise intervention was also effective at improving the lung volumes in asthmatic children. However, the quantum of reduction in lung obstruction and consequential overall improvement in lung functions were found to be more significant with the aerobic exercise intervention than breathing exercise intervention only. Similarly, in our study, breathing exercises alone appear to be partially effective but less effective compared to a combination of aerobic-breathing exercises in supportive asthma treatment.

We can assume that untrained patients in experimental group 2 achieved a certain degree of increased contractility and hypertrophy of the breath pump muscles due to adaptation to aerobic activity. In the research, we were unable to determine the effects of the breathing exercises and aerobic program on significantly higher changes in the FVC parameter compared to the control group. The reason for this fact may be the number of determinants that disrupted the course of research (genetic predisposition), but also the

choice of the nature of the experimental stimuli. We assumed that an aerobic program with different intensity or duration is needed to achieve more significant changes.

This study suggests that changes in spirometry parameters in patients with bronchial asthma can also be achieved by simple breathing and aerobic exercises, the application of which is possible anywhere and without any special aids. Therefore, in addition to the treatments prescribed by a doctor, we consider it beneficial to use various proven methods of breathing exercises, aerobic programs, and various specialized training programs that can improve the patient's current condition.

Some major limitations of this study are that it involved a small group of patients and only women at similar age. For groups in this experiment, a more detailed medical record would be more appropriate. For small samples, statistical significance may not be apparent. The next limitation was the duration of the experiment. A longer experimental period could cause more significant changes in the monitored parameters. The length and frequency of stimuli in the individual groups were not exactly the same. Although the results suggested high efficiency of experimental stimuli, they cannot be generalized for the mentioned work limits.

## 5. Conclusions

The aim of the study was to investigate the effects of breathing exercises and a combination of breathing exercises and aerobic program on changes in basic spirometry parameters in standardized diagnostics in patients with bronchial asthma in the presence of a clinical immunologist. In the research work, we applied a three-group time-parallel experiment in which we monitored selected spirometry parameters using a standardized spirometer. In each separate patient group, we observed significant improvements in all parameters examined when comparing pre-training and post-training diagnostics. Comparing the post-training values of the individual groups, we found that participants in experimental group 1, which performed breathing exercises, significantly improved in FEV1 parameters and in FEV1/FVC compared to the control group. Patients in experimental group 2, who performed a combination of breathing exercises with an aerobic program, significantly improved in the parameters FEV1, FEV1/FVC, PEF and FEF25–75 compared to the control group. Based on these facts, we can state that when patients with bronchial asthma perform breathing exercises, there is a significant positive effect on improving their lung function in the aforementioned parameters. However, the application of a program combining breathing exercises with an aerobic program seems to be an option that brings significant improvements in spirometry parameters. When comparing the effects of experimental factors, we did not find significant differences between the groups. The reason for this finding may be the nature of aerobic activity, which was performed at higher pulse values and contained higher frequencies of breath cycles than breathing exercises, which allowed patients to train in a more intense flow zone by walking or running.

**Author Contributions:** Conceptualization, Ľ.G., D.S. and I.M.; methodology, Ľ.G. and J.L.; investigation, D.S.; data curation, L.O.; writing—original draft preparation, L.O.; writing—review and editing, I.M., Ľ.G., J.L. and L.O.; visualization, L.O.; supervision, Ľ.G. and J.L. All authors have read and agreed to the published version of the manuscript.

**Funding:** This work was supported by the Scientific Grant Agency of the Ministry of Education, Science, Research and Sport of the Slovak Republic and the Slovak Academy of Sciences (Erika Zemková: 1/0089/20).

**Institutional Review Board Statement:** All protocols were approved by the Ethics Committee of the Faculty of Physical Education and Sport, Comenius University in Bratislava (number 1/2020). Participants provided written informed consent.

**Informed Consent Statement:** Informed consent was obtained from all subjects involved in the study.

**Data Availability Statement:** The data presented in this study are available on request from the corresponding author. The data are not publicly available due to ethical and privacy restrictions.

**Conflicts of Interest:** The authors declare no conflict of interest.

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
