# Peer review of "Influences of Breathing Exercises and Breathing Exercise Combined with Aerobic Exercise on Changes in Basic Spirometry Parameters in Patients with Bronchial Asthma"

_applsci, doi:10.3390/app12147352_

Round 1
Reviewer 1 Report
Interesting idea of ​​your study, my recommendations are the following:
In the abstract, I recommend that when mentioning the results, put a full stop and not a line, even if you mentioned parametrically that they improve, so as not to create confusion. I recommend reformulating the conclusion, more concise and clear, e.g. appears to be….
Experimental protocol 2 - according to the mentions, 80% physical activity is performed, which means that at the age of 34, the maximum value is around 160 BPM. The indication is erroneous, I recommend to mention the threshold that should not be exceeded by 80% of 160. Because in the mentioned conditions you are at a maximum intensity.
Line 158 mention only one exercise, which is That, I recommend the description. I think it is a set of exercises, performed for 30 minutes, I recommend clarification. I also recommend inserting some examples of exercises mentioning the dosage as in the first protocol.
Line 159, I recommend replacing the words cool down with exercises on the body's recovery after exertion.
Line 249-251 to which authors you refer, I recommend clarification. I recommend reformulating the whole sentence, because various ventilations probably refer to the level of respiratory capacity.
Line 254-257 I recommend deleting, it is enough to mention the purpose, and the idea is repeated in the introduction.
Discussions - I recommend to make more correlations between the concrete results of this study with results from other studies, not only in terms of theory but also in terms of value, mathematics.
Author Response
Point 1: In the abstract, I recommend that when mentioning the results, put a full stop and not a line, even if you mentioned parametrically that they improve, so as not to create confusion. I recommend reformulating the conclusion, more concise and clear, e.g. appears to be…
Response 1: Diacritics – fixed
Conclusion – reformulated
Point 2: Experimental protocol 2 - according to the mentions, 80% physical activity is performed, which means that at the age of 34, the maximum value is around 160 BPM. The indication is erroneous, I recommend to mention the threshold that should not be exceeded by 80% of 160. Because in the mentioned conditions you are at a maximum intensity.
Response 2: There was a misunderstanding about 160 BPM. I fixed it and explained.
Maximal heart rate was calculated as result of formula 220 – age. Probands was instructed to maintain at 80 % of HR max (± 2 BPM) and not to exceed this threshold.
Point 3: Line 158 mention only one exercise, which is That, I recommend the description. I think it is a set of exercises, performed for 30 minutes, I recommend clarification. I also recommend inserting some examples of exercises mentioning the dosage as in the first protocol.
Response 3: For each proband, the exercise was chosen individually according to his current physical condition. Probands could choose running or Nordic walking. It was fixed in manuscript.
Point 4: Line 159, I recommend replacing the words cool down with exercises on the body's recovery after exertion.
Response 4: Replaced
Point 5: Line 249-251 to which authors you refer, I recommend clarification. I recommend reformulating the whole sentence, because various ventilations probably refer to the level of respiratory capacity.
Response 5: reformulated and fixed
Point 6: Line 254-257 I recommend deleting, it is enough to mention the purpose, and the idea is repeated in the introduction.
Response 6: deleted
Point 7:Discussions - I recommend to make more correlations between the concrete results of this study with results from other studies, not only in terms of theory but also in terms of value, mathematics.
Response 7: Added 2 meta-analysis with statistical and mathematical results.

Reviewer 2 Report
Although the study design is adequate and the presentation of the results is good, the discussion and conclusion do not follow the effect size and statistical significance found.
I feel that the authors would like to find an effect of the interventions that would outperform the control group. But I am not conviced tha aerobic or breathing exercise are much better than control in this study. I am also not conviced that breathing exercise alone is not as effective as aerobic exercise with breathing exercise. The discussion is not convincing as it seems inconsistent with the results.
I recommend major adjustments in the discussion and conclusion. Please, find the article attached with other specific comments.

Author Response
- All corrections and inconsistencies in the text have been corrected and highlighted.
- Two meta-analyses were added to the discussion, with which we compared our results.
- The main shortcoming of the work was the lack of statistical comparison of groups in the results section. Statistical comparison has been added in results section and highlighted. We realize that without this comparison, the discussion and conclusion could have seemed confused. Several changes and corrections have been made also in discussion and conclusion section.

Round 2
Reviewer 1 Report
No comments
Author Response
Thank you for your review.
Reviewer 2 Report
I would recommend including statistical significance in the figures for post-treatment comparisons between the control, EX1 and EX2 groups. I.e., including a symbol like "*" for significant p. The authors clarified this issue in "Results", but the reader could benefit from the same clarification in the figures.
Author Response
Statistical significance was add in the figures.
Thank you for your review.